# Prognostic and clinicopathological significance of GPRC5A in various cancers: A systematic review and meta-analysis

**Lu Dai, Xiao Jin, Zheng Liu** ⬛ *

Medical Centre for Digestive Diseases, Second Affiliated Hospital, Nanjing Medical University, Nanjing, Jiangsu Province, People's Republic of China

* liuzheng117@njmu.edu.cn

**Data Availability Statement:** All relevant data are within the paper and its Supporting Information files.

**Funding:** The authors received no specific funding for this work.

## Abstract

### Background

GPRC5A is associated with various cancer initiation and progression. Controversial findings have been reported about GPRC5A prognostic characteristics, and no meta-analysis has been conducted to assess the relationship between GPRC5A and cancer prognosis. Therefore, the objective of this meta-analysis is to evaluate the overall prognostic effectiveness of GPRC5A.

### Methods

We first conducted a systematic search in the PubMed, Embase, Web of Science, CNKI, Cochrane, and WangFang databases. The hazard ratio (HR) and odds ratios (OR) with 95% CI were then pooled to assess the associations between GPRC5A expression and overall survival (OS), disease-free survival (DFS), event-free survival (EFS), and clinicopathological characteristics. Chi-squared test and $I^2$ statistics were completed to evaluate the heterogeneity in our study. A random-effects model was used when significant heterogeneity existed ($I^2$>50% and p<0.05); otherwise, we chose the fixed-effect model. Subgroup analysis was stratified by tumor type, region, HR obtained measurements, and sample capacity to explore the source of heterogeneity.

### Results

In total, 15 studies with 624 patients met inclusion criteria of this study. Our results showed that higher expression of GPRC5A is associated with worse OS (HR:1.69 95%CI: 1.20–2.38 $I^2$ = 75.6% $p$ = 0.000), as well as worse EFS (HR:1.45 95%CI: 1.02–1.95 $I^2$ = 0.0% $p$ = 0.354). Subgroup analysis indicated that tumor type might be the source of high heterogeneity. Additionally, cancer patients with enhanced GPRC5A expression were more likely to lymph node metastasis (OR:1.95, 95%CI 1.33–2.86, $I^2$ = 43.9%, $p$ = 0.129) and advanced tumor stage (OR: 1.83, 95%CI 1.15–2.92, $I^2$ = 61.3%, $p$ = 0.035), but not associated with age, sex, differentiation, and distant metastasis.

**Competing interests:** The authors have declared that no competing interests exist.

**Abbreviations:** GPRC5A, G-protein-couple receptor, family C, group 5 member A; RAI3, retinoic acid-induced protein 3; RAIG1, retinoic acid-inducible gene 1; GPCRs, G protein-coupled receptors; HRs, hazard ratios; ORs, odds ratios; CIs, confidence intervals; OS, overall survival; DFS, disease-free survival; EFS, event-free survival; NOS, Newcastle-Ottawa Scale; IHC, Immunohistochemical.

## Conclusion

GPRC5A can be a promising candidate for predicting medical outcomes and used for accurate diagnosis, prognosis prediction for patients with cancer; however, the predictive value of GPRC5A varies significantly according to cancer type. Further studies for this mechanism will be necessary to reveal novel insights into application of GPRC5A in cancers.

## Introduction

Cancer is an important public health issue worldwide, and most cancer-associated deaths (90%) are caused by metastatic cancer [1]. Due to most patients are already at the advanced stage when diagnosed, which makes them have little chance to be treated with surgery or radiotherapy It is crucial to identify diagnostic or prognostic biomarkers and therapeutic targets to improve clinical outcomes. Cancer molecular targeted therapy has recently experienced remarkable advances. The targeted drugs approved by the FDA for clinical use in the last decades, such as nivolumab [2], olaparib [3], and blinatumomab [4], have made substantial improvements in cancer diagnosis and therapy, thus producing optimism in the fight against cancer. Although many tumor markers have been found playing an essential role in various cancers in recent years, only a few of them have been used in clinical practice.

G-protein-couple receptor, family C, group 5 member A (GPRC5A), which was identified as an all-*trans*-retinoic acid-inducible protein, is a member of class c orphan GPCRs. GPRC5A, also known as RAI3 or RAIG1, can activate numerous signal transduction cascades, including the NF-κB, cAMP-Gs α, FAK/Scr, and STAT3 signaling pathways [5]. Recently, GPRC5A has been proven to be an essential role in human cancers. However, the biological functions of GPRC5A are controversial. Previous research indicated that GPRC5A is considered an anti-oncogene in lung cancer and the expression level of GPRC5A in lung cancer tissue is much lower than that in normal lung tissue [6]. Tao Q et al. reported that high expression of GPRC5A inhibited NSCLC cell viability and enhanced apoptosis in an in vitro experiment [7]. Later, the same tumor suppressor ability was found in head and neck squamous cell carcinoma (HNSCC) [8] and oral squamous cell carcinoma (OSCC) [9]. Dysregulation of GPRC5A expression has been observed in other human cancers, such as breast cancer [10], colorectal cancer [11], gastric cancer [12], hepatocellular carcinoma [13], pancreatic cancer [14], prostate cancer [15] and ovarian cancer [16]. However, GPRC5A was also found to serve as an oncogene in these cancers, and the high expression of GPRC5A was related to tumorigenesis. This dual behavior makes GPRC5A a fascinating gene to study.

The prognostic role of GPRC5A in cancer remains unclear. Jin E et al. found that lung cancer patients with high expression of GPRC5A tended to have a better prognosis [17]. Similarly, several studies suggest that overexpression of GPRC5A predicts poor prognosis in gastric [14], pancreatic [16], and colorectal cancer [12] patients. Moreover, some contradictory views exist about the prognostic effect of GPRC5A in various cancers, including breast cancer [10], pancreatic cancer [14], and hepatocellular cancer [13]. Nagahata et al. [10] implied that the upregulation of GPRC5A might be a frequent feature of poor prognosis in breast cancer. However, Cheng et al. reported that there was no significant correlation between GPRC5A expression and clinicopathological characteristic immunohistochemical (IHC) tissue microarray analysis [18]. Therefore, the results of different studies are controversial and no meta-analysis has been performed to assess the diagnostic utility of GPRC5A among multiple types of cancer. In response, this meta-analysis was conducted to examine the prognostic value of GPRC5A in different types of human cancers from a collection of published results.

## Methods

### Search strategy

This study followed the PRISMA (Preferred Reporting Items for Systematic Reviews and Meta-Analyses) guidelines [19], we conducted a systematic search of PubMed, Embase, Web of Science, CNKI, Cochrane Library, and WangFang on July 8, 2020, using the MeSH terms. The studies were identified using the search strategy: (GPRC5A OR RAI3 OR G protein-coupled receptor, family C, group 5, member a protein OR RAIG1) AND (cancer OR carcinoma OR tumor OR neoplasm OR malignancy). All English and Chinese studies were enrolled. Reference lists of the included articles were reviewed to get all the reports about GPRC5A before the deadline.

### Inclusion and exclusion criteria

Inclusion criteria include (1) studies of adult human; (2) the cancers in the studies must be certified by the gold standard; (3) GPRC5A was tested by immunohistochemistry(IHC); (4) the content of the article focus on the correction of GPRC5A with the prognosis and clinicopathologic characteristics of multiple cancers; (5) hazard ratio(HR) and 95% confidence interval(CI) were provided in article or enough data was given to calculate the HR with 95% CI; (6) all English and Chinese studies were included.

Exclusion criteria include (1) reviews or meta-analysis; (2) letters, conference reports and basic studies; (2) duplicated records; (3) no sufficient data to calculate the HR with 95%CI.

### Data extraction and quality assessment

Two investigators separately gained the information and data from primary publications. The specific information and data were as follows: the first author's name, publication year, country, cancer type, time of sample collection, sample capacity, outcome measures, method of detection, and cut-off value. For the clinically relevant factors, age, sex, differentiation, tumor invasion depth, lymph node metastasis, and distant metastasis were extracted.

The Newcastle-Ottawa Scale (NOS) was also utilized to assess the quality of studies in the meta-analysis, which ranges from 0–9. A score of 5 or higher indicates strong evidence; a score from 4 to 5 (not included) indicates medium evidence, and a score below 4 indicates weak evidence. Studies with strong evidence (NOS score $\geq$ 5 points) were included in this study [20].

### Statistical analysis

We used Stata 15.1 software to calculate extracted data in the meta-analysis, and Engauge Digitizer 10.0 was used to get survival data when literature only provided a Kaplan-Meier curve. Pooled HRs with 95% CI for OS, DFS, EFS, and odds ratios (ORs) for clinicopathological parameters were calculated. If HR or OR>1, and 95%CI did not contain 1, the study was recognized as statistically significant. Chi-squared test and $I^2$ statistics were completed to evaluate the heterogeneity in our study. A random-effects model was used when significant heterogeneity existed ($I^2$>50% and p<0.05); otherwise, we choose the fixed-effect model. Subgroup analysis was stratified by tumor type, region, HR obtained measurements, and sample capacity to explore the source of heterogeneity.

Publication bias was assessed by Begg's and Egger's tests [21]. Meanwhile, a sensitivity analysis was performed to assess the reliability and stability of our results; *p* values<0.05 meant statistically significant.

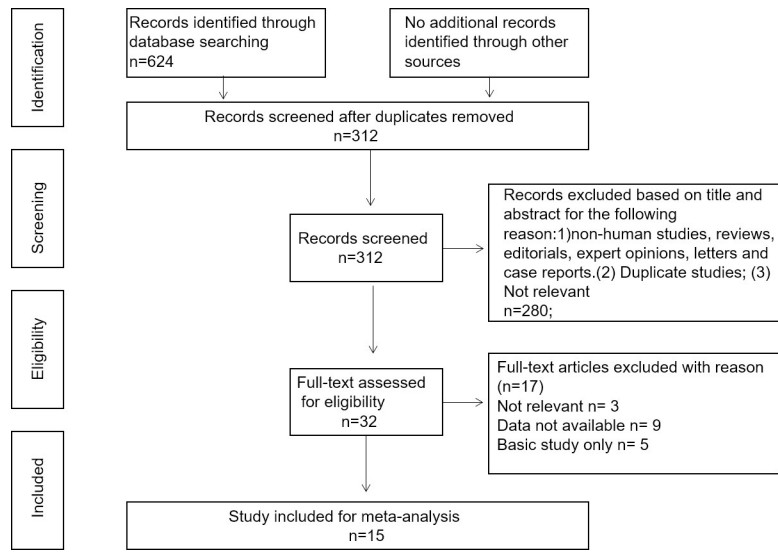

**Fig 1. Flow diagram of the study search and selection process.**

## Results

### Search results and study characteristics

According to the literature selection procedure (Fig 1), 624 studies related to GPRC5A and cancer prognosis were identified from online database searches using specific search terms. A total of 609 articles were excluded after reviewing the titles, abstracts and data. Finally, 15 studies comprising a total of 3349 patients were included in the meta-analysis. The types of cancer in the studies included: colorectal cancer [22, 23], pancreatic cancer [24–27], gastric cancer [28, 29], esophageal cancer [30], prostate cancer [31], hepatocellular cancer [32], head and neck squamous cell carcinoma [33], lung cancer [34], and ovarian cancer [35]. The characteristics of the enrolled articles are summarized in Table 1. All studies were published from 2013

**Table 1. The main characteristic of enrolled records in the meta-analysis.**

| Author | Year | Country | Cancer type | Included period | Sample | Source of HR | Endpoint | Assay method | Cut-off value | NOS |
|---|---|---|---|---|---|---|---|---|---|---|
| Greenhough et al | 2018 | UK | CRC | N | 320 | indirectly | EFS | N | N | 7 |
| Jahny et al | 2016 | Germany | PDAC | N | 376 | indirectly | OS | IHC | staining intensist>30% | 6 |
| Liu et al | 2015 | China | GC | 2005–2009 | 106 | directly | OS | IHC | Score≥6 | 6 |
| El et al | 2019 | Germany | ESCC | N | 235 | indirectly | OS | IHC | staining intensist>2+ | 8 |
| Sawada et al | 2019 | Japan | PCa | N | 421 | directly | 0S | N | N | 7 |
| Zheng et al | 2013 | China | HCC | 2001–2009 | 106 | directly | OS/DFS | IHC | Score>4 | 6 |
| Liu et al | 2017 | China | HNSC | N | 86 | indirectly | OS/DFS | IHC | Score>30% | 6 |
| Zougman et al | 2013 | UK | CRC | N | 367 | directly | OS | IHC | Score>3 | 5 |
| Jin et al | 2018 | China | LC | 2007–2009 | 110 | indirectly | OS | IHC | Score≥6 | 7 |
| Galceran et al | 2019 | Sweden | OC | 2002–2006 | 136 | indirectly | OS/EFS | IHC | Score>2 | 6 |
| Er et al | 2020 | China | LC | N | 503 | indirectly | OS | IHC | N | 6 |
| Melling et al | 2019 | Germany | GC | 1994–2006 | 98 | indirectly | OS | IHC | Score≥2 | 7 |
| Chang et al | 2019 | China | PDAC | N | 135 | indirectly | OS | N | N | 6 |
| Jiang et al | 2019 | China | PC | N | 176 | indirectly | OS/DFS | N | N | 8 |
| Wu et al | 2019 | China | PC | N | 174 | indirectly | OS | N | N | 6 |

to 2020, and the sample sizes ranged from 86 to 503 patients. The expression level of GPRC5A was detected by immunohistochemistry (IHC), and patients were divided into two groups according to their GPRC5A expression level. Of the eligible articles, eight studies were from China, three were from Germany, two were from the UK, one was from Japan, and one was from Sweden. Of these 15 studies, 14 were used to evaluate the HR of OS, three were used to evaluate the HR of DFS, and two were used to evaluate the HR of EFS. Moreover, the NOS score varied from five to eight, suggesting that our study was of solid methodological quality.

## Association between GPRC5A level and OS

In this meta-analysis, fourteen studies demonstrated the correction between GPRC5A expression and OS in human cancer pooled HRs with 95%CIs evaluated for OS was (1.69 95%CI 1.20–2.38 p = 0.05), using a random-effects model for high heterogeneity ($I^2$ = 75.6% P< 0.001) (Fig 2). These results indicated that GPRC5A might be a predictive factor for tumor

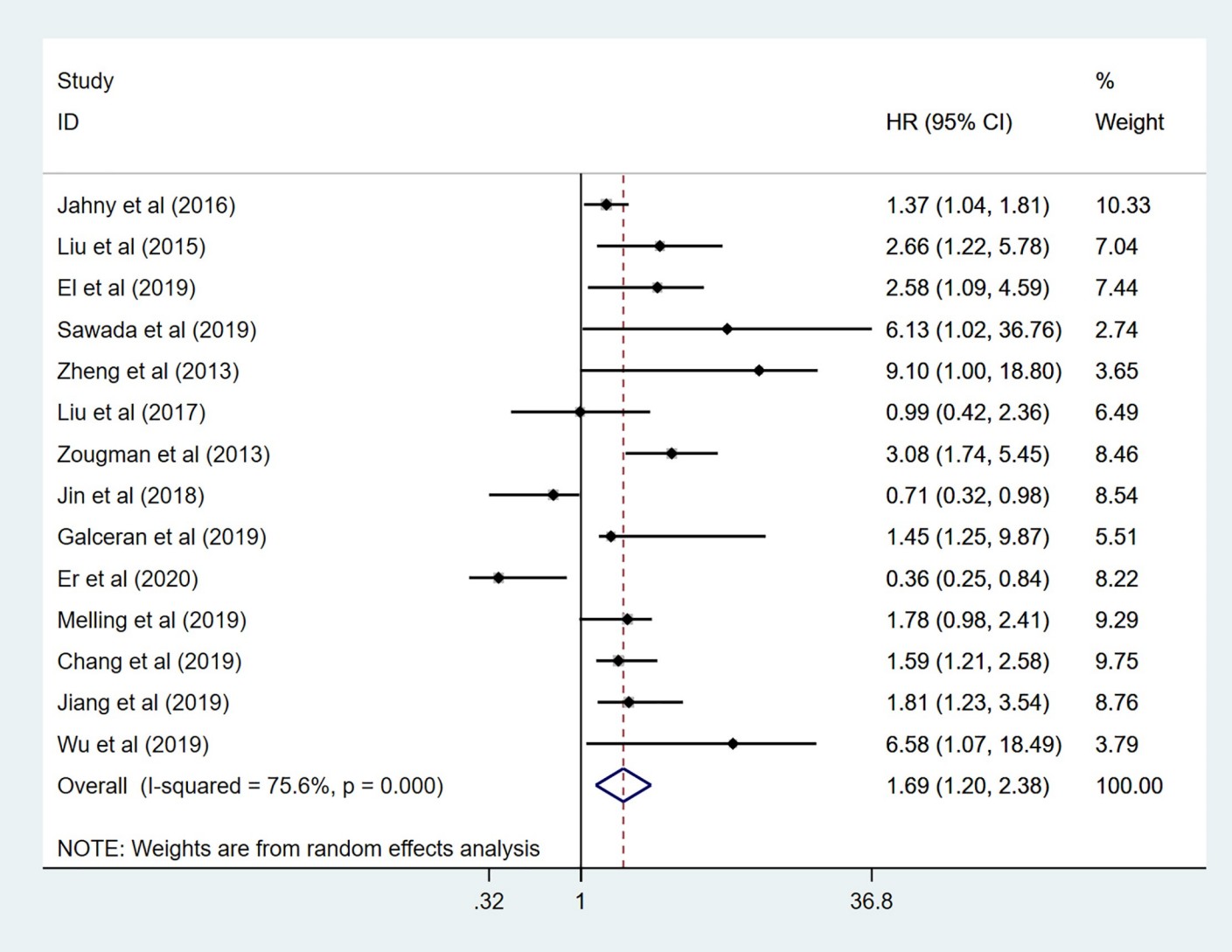

**Fig 2. Forest plot for the association between GPRC5A expression and OS.**

prognosis and high GPRC5A density is associated with a significantly lower OS rate in cancer patients.

## Association between GPRC5A level and DFS and EFS

Three records, including 368 patients, evaluated HRs for DFS. There was statistically significant heterogeneity ($I^2$ = 73.6% $p$ = 0.023); thus, a random-effect model was conducted to calculate the pooled HR for DFS: (1.86 95%CI 0.70–4.91, $p$ = 0.05) (Fig 3). We found that there was no significant association of high GPRC5A expression with lower DFS. When we come to EFS, only two studies provided data to calculate the HRs. As shown in Fig 4, results showed that high GPRC5A expression was significantly related to pooler EFS in this meta-analysis (HR:1.41 95%CI 1.02–1.95 $I^2$ = 0.0% $p$ = 0.354).

## Subgroup analysis of the prognostic effect of GPRC5A among multiple cancer

To explore the sources of high heterogeneity in this meta-analysis, subgroup analyses for OS data were performed by tumor type, region, HR obtained measurements, sample capacity (Table 2). Among 14 enrolled studies in the subgroup analysis, nine different cancers have been discussed, including pancreatic cancer (n = 4), gastric cancer (n = 2), lung cancer (n = 2), prostate cancer (n = 1), hepatocellular cancer (n = 1), head and neck squamous cell carcinoma

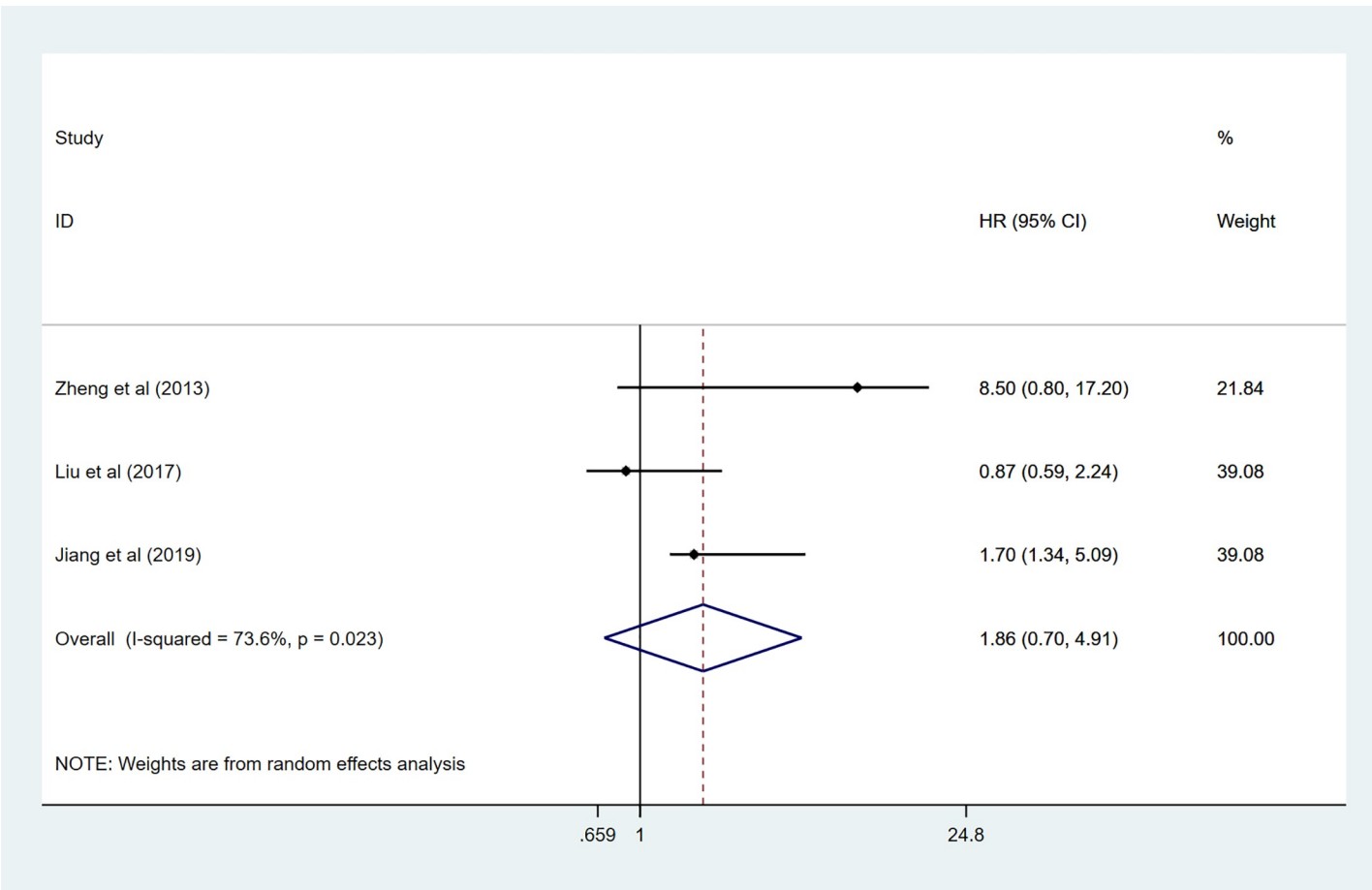

**Fig 3. Forest plot for the association between GPRC5A expression and DFS.**

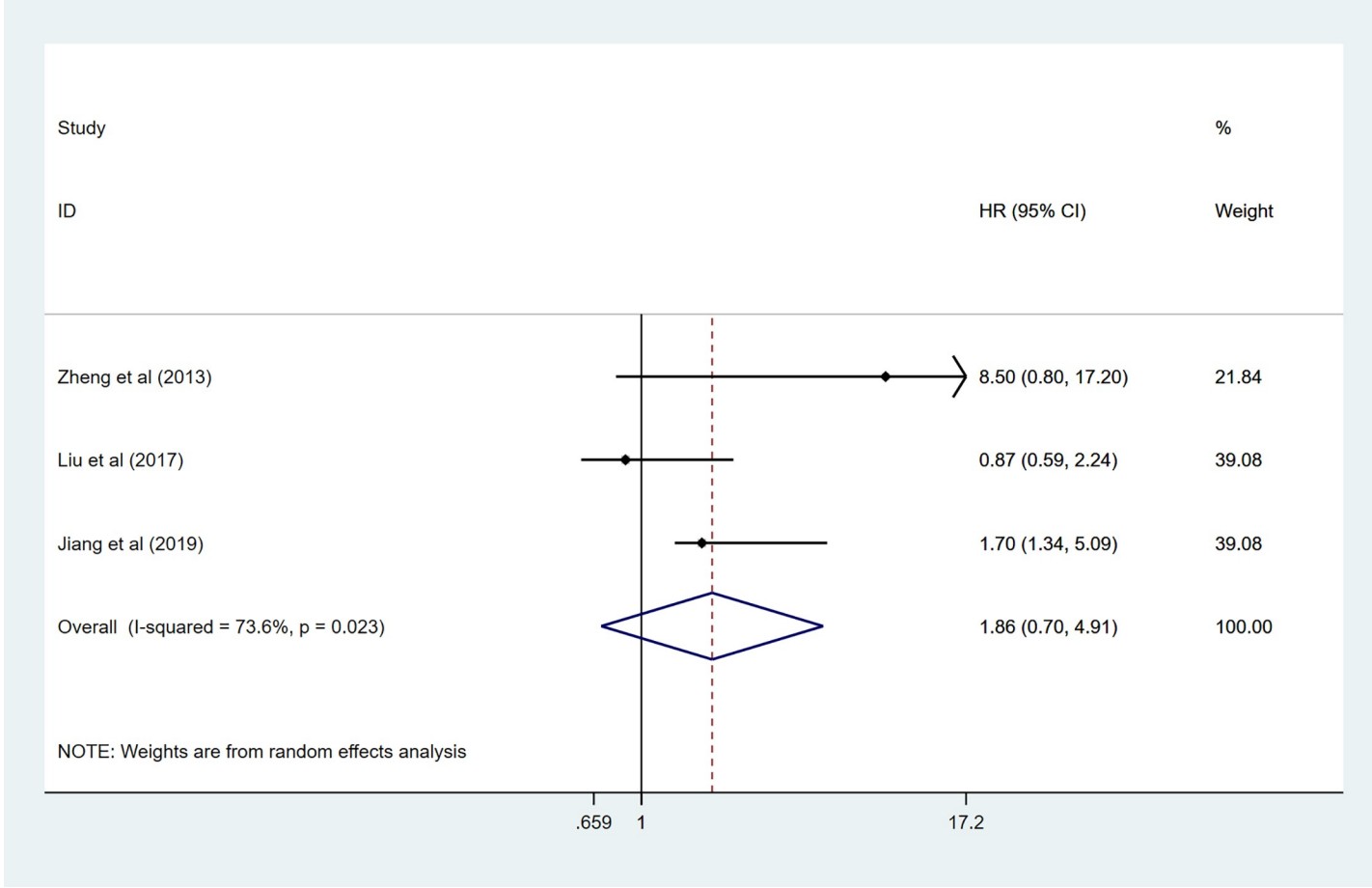

**Fig 4. Forest plot for the association between GPRC5A expression and EFS.**

**Table 2. Subgroup analysis for the correction between GPRC5A and OS.**

| Variables | Studies(n) | HR 95%CI | Heterogeneity | | Model |
|---|---|---|---|---|---|
| | | | I2 | P | |
| Cancer type | | | | | |
| Digestive cancer | 4 | 2.32(1.73–3.11) | 0.0% | 0.479 | Random effects |
| Non-digestive cancer | 10 | 1.29(1.09–1.54) | 77.2% | 0.000 | Random effects |
| Lung cancer | 2 | 0.52(0.34–0.78) | 61.6% | 0.107 | Random effects |
| Other cancers except lung cancer | 12 | 1.77(1.50–2.07) | 47.9% | 0.032 | Random effects |
| Region | | | | | |
| Asian | 9 | 1.31(1.05–1.63) | 81.2% | 0.000 | Random effects |
| Non-Asian | 5 | 1.69(1.38–2.07) | 49.6% | 0.094 | Random effects |
| HR obtained method | | | | | |
| Direct method | 4 | 3.36(2.19–5.14) | 0.0% | 0.450 | Random effects |
| Indirect method | 10 | 1.34(1.14–1.57) | 74.3% | 0.000 | Random effects |
| Sample size | | | | | |
| n>150 | 7 | 1.50(1.23–1.83) | 83.8% | 0.000 | Random effects |
| n<150 | 7 | 1.50(1.20–1.89) | 63.2% | 0.012 | Random effects |

(n = 1), colorectal cancer (n = 1), ovarian cancer (n = 1), and esophageal cancer (n = 1). As been shown in Fig 5, the mixed effects of individual cancers have been calculated. High expression level of GPRC5A predicted a worse OS in pancreatic cancer (HR: 1.71 95%CI: 1.29–2.26 $I^2 = 23.8\%$ $p = 0.268$), Gastric cancer (HR: 1.97 95%CI: 1.33–2.91 $I^2 = 0.0\%$ $p = 0.381$), prostate cancer (HR: 6.13 95%CI:1.02–36.80), hepatocellular cancer (HR: 9.10 95%CI: 2.10–39.46), and esophageal cancer (HR:2.58 95%CI:1.26–5.29). However, there was no association in colorectal cancer (HR:1.40 95%CI:0.73–2.66), ovarian cancer (HR:1.45 95%CI:0.52–4.07), and head and neck squamous cell carcinoma (HR: 0.87 95%CI:0.45–1.70). Controversial results observed in lung cancer with pooled HR = 0.73 (95%CI: 0.49–1.11 $I^2 = 0.0\%$ $p = 0.872$) and subgroup analysis also showed GPRC5A expression predicted unfavorable prognosis in digestive system (HR: 2.32 95%CI: 1.73–3.11 $I^2 = 0.0\%$ $p = 0.479$) (Fig 6), and other systems (HR: 1.29 95% CI:1.09–1.54 $I^2 = 77.2\%$ $p<0.000$). To further explore the sources of high heterogeneity, a sensitivity analysis was performed. As shown in Fig 7, the observed heterogeneity disappeared while omitting the two lung cancer studies as a source of heterogeneity.

In addition, to further demonstrate the predictive role of GPRC5A in multiple tumors, more subgroup analysis was conducted based on region, HR obtained measurements, sample capacity. When it came to the region, pooled HR was (1.31 95%CI 1.05–1.63) for the Asian records with high heterogeneity ($I^2 = 81.2\%$, $p<0.001$). as for the non-Asian subgroup, pooled HR was (HR:1.69, 95%CI 1.38–2.07) with no heterogeneity ($I^2 = 49.6\%$ $p = 0.094$) (Fig 8). The combined HR for large and small sample capacity were 1.50 (95%CI 1.23–1.83 $I^2 = 83.8\%$ $p<0.001$) and 1.50 (95%CI 1.20–1.89 $I^2 = 63.2\%$ $p = 0.012$) (Fig 9). In our meta-analysis, only a few studies provided specific HR data evaluated for OS. Indirectly data based on the Engauge Digitizer could contribute to the high heterogeneity. The pooled HR for directly and indirectly HR obtained measurement were 3.36 (95%CI 2.19–5.14 $I^2 = 0.0\%$ P = 0.450) and 1.34 (95%CI 1.14–1.57 $I^2 = 74.3\%$ P = 0.000) (Fig 10). Therefore, these results of subgroup analysis suggested that tumor type might be a major source of high heterogeneity.

### Association between GPRC5A level and clinicopathological parameters

Due to the limitation of the data, we only analyzed clinicopathologic features including age, sex, differentiation, tumor invasion depth, lymph node metastasis, and distant metastasis. All the detailed data are summarized in Table 3. High expression of GPRC5A was associated with Lymph node metastasis (OR:1.95, 95%CI 1.33–2.86, $I^2 = 43.9\%$, $p = 0.129$), advanced Tumor stage (OR: 1.83, 95%CI 1.15–2.92, $I^2 = 61.3\%$, $p = 0.035$), but not related to age (OR:1.14, 95% CI 0.86–1.52, $I^2 = 0.0\%$, $p = 0.578$), sex (OR 1.17, 95%CI 0.91–1.50, $I^2 = 0.0\%$, $p = 0.863$), differentiation (OR: 1.58, 95%CI 0.71–3.50, $I^2 = 88.5\%$, $p<0.001$), and distant metastasis (OR 1.40,95%CI0.59-3-32, $I^2 = 65.0\%$, $p = 0.036$).

### Sensitivity analyses and publication bias

A sensitivity analysis was conducted to evaluate the reliability and stability of our results by omitting any individual studies (Fig 11). Fortunately, the pooled HR for OS was not influenced; this suggested that the result of our meta-analysis was believable. Additionally, Begg's funnel plots and Egger's test showed no significant publication bias was found for OS (Fig 12 Begg'sP ¼ 0.155 and Egger's P ¼ 0.908).

## Discussion

The lack of effective biomarkers for early diagnosis of aggressive cancers is one of the most intractable issues in clinical cancer management. Recently, GPRC5A has received attention for its intriguing dual behavior in cancer. GPRC5A is dysregulated in several human cancers and

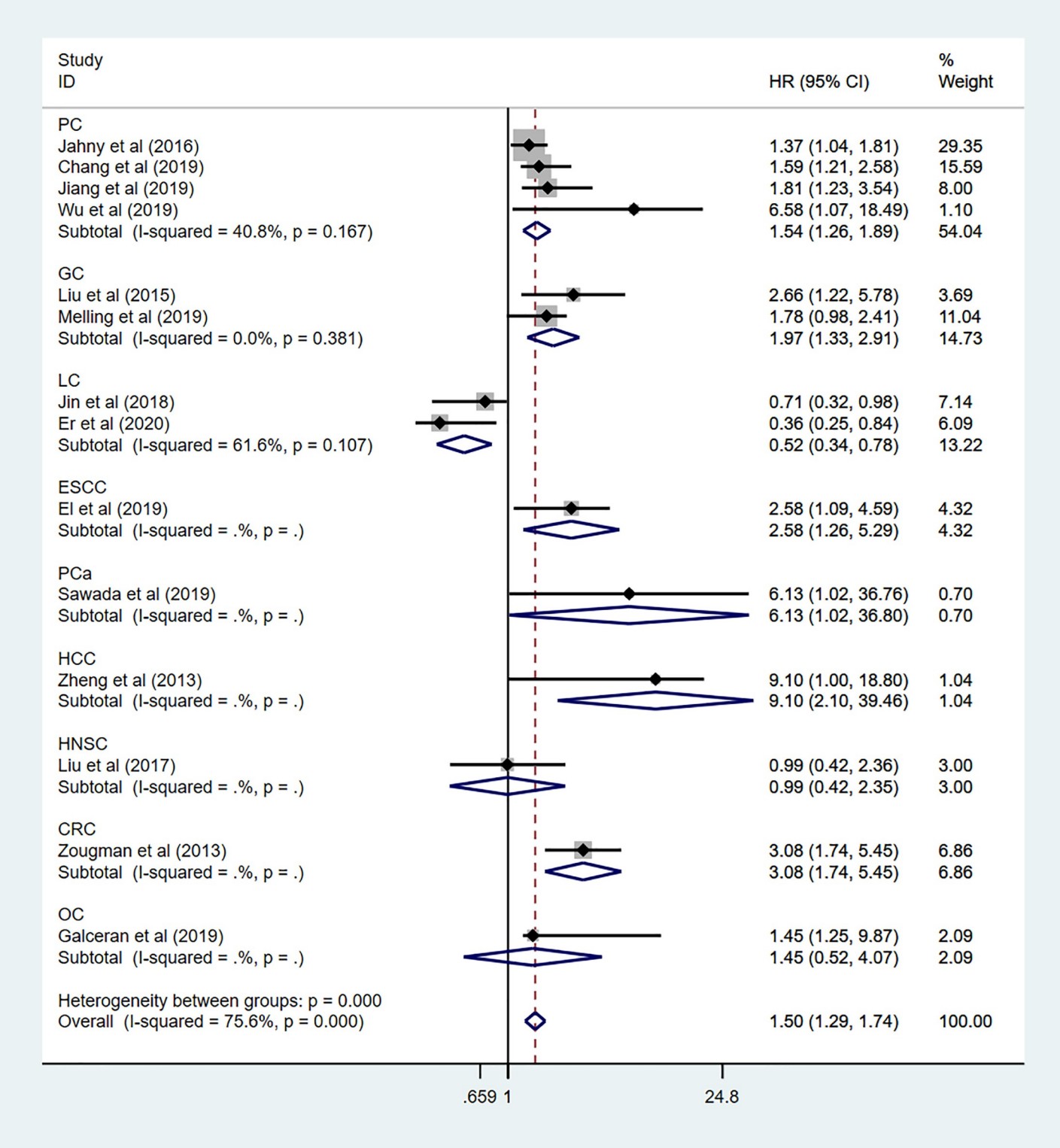

**Fig 5. Forest plot of Overall analysis of GPRC5A expression in multiple carcinoma.**

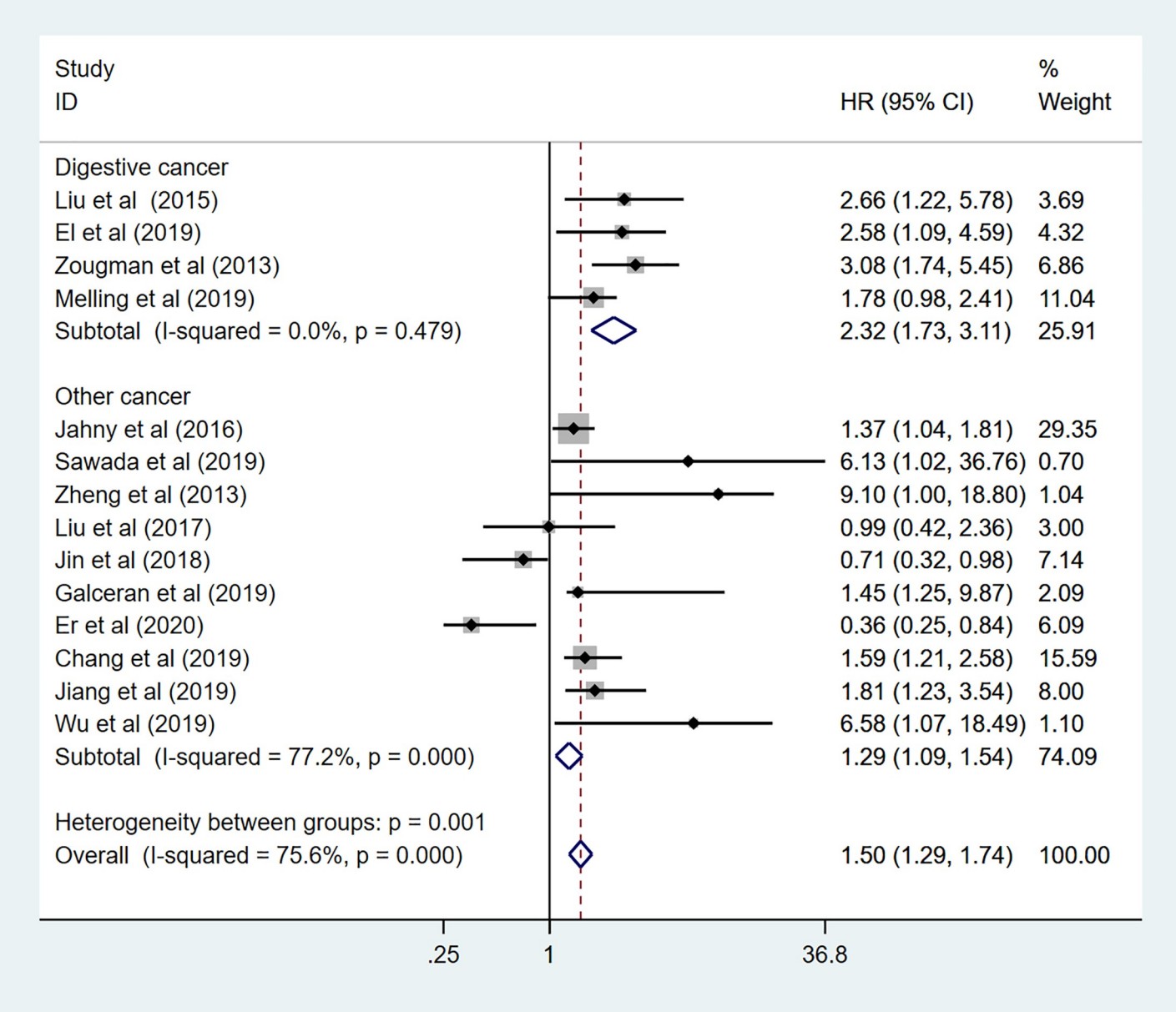

**Fig 6. Forest plot for the subgroup analysis.** Digestive cancer.

may be a candidate target for cancer treatment. The effect of GPRC5A in various cancers has been reported to be different. GPRC5A has been reported as a tumor-suppressor gene in lung cancer. Loss of the GPRC5A gene may lead to the development of spontaneous lung cancers in mice [36]. Fujimoto J et al. found that deletion of GPRC5A can promote tumorigenesis by activating the NF-κB signaling pathway, which may lead to the development of acidophilic macrophage pneumonia (AMP) in GPRC5A knockout mice [37]. However, GPRC5A frequently expresses oncogenic characteristics in other cancers such as colon, gastric, liver, breast and pancreatic cancers. Knockdown of GPRC5A with siRNA can promote tumor cell apoptosis and reduce cell proliferation in colorectal cancer [38]. A vivo mouse model found that a lack of GPRC5A inhibited colitis-associated tumorigenesis [39]. In gastric cancer, GPRC5A

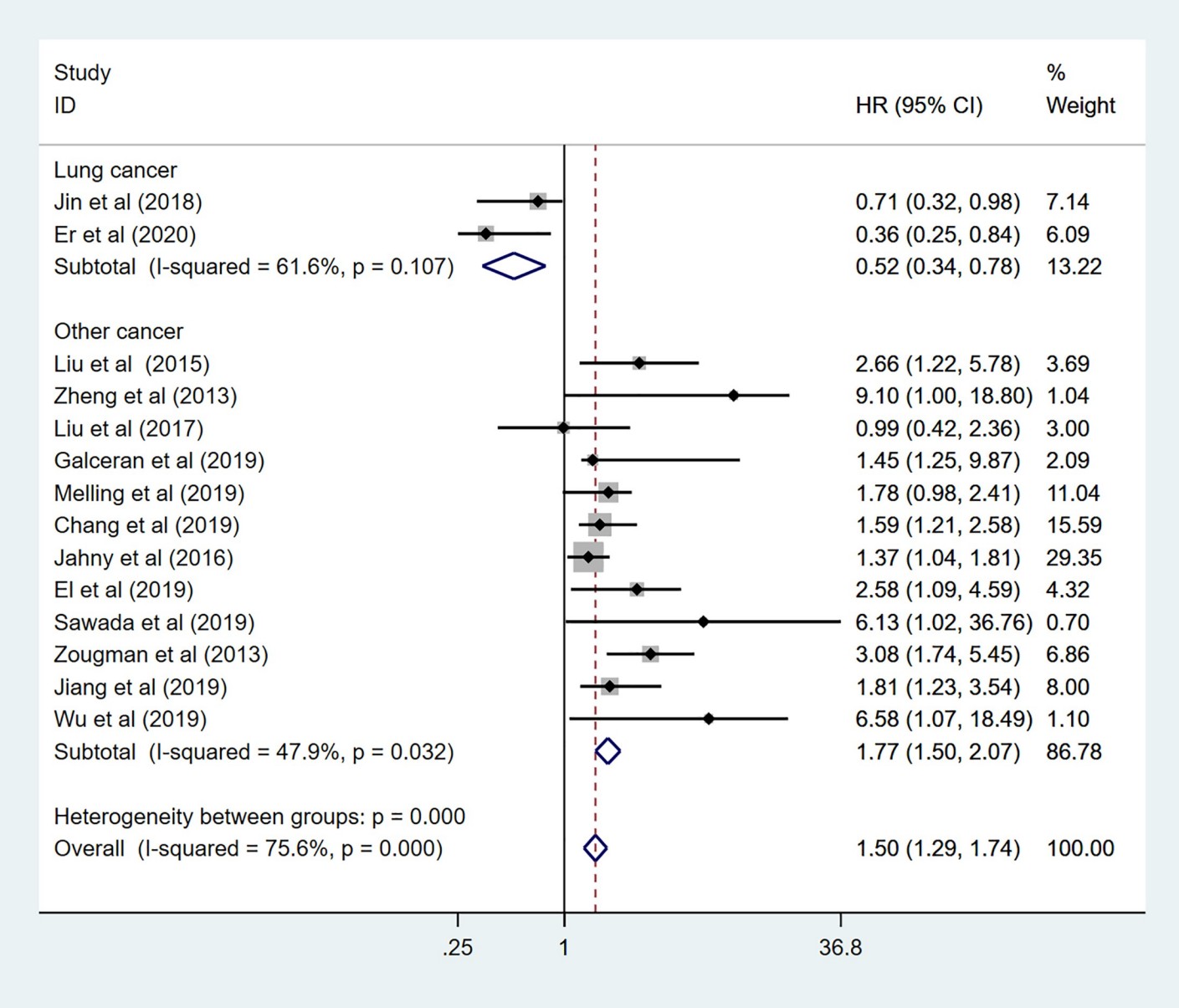

**Fig 7. Forest plot for the subgroup analysis.** Non-Lung cancer.

expression levels are elevated in GC tissues compared with normal tissues and high expression of GPRC5A is significantly related to aggressive clinicopathological parameters and poor OS [40]. The same oncogenic characteristics of GPRC5A were observed in HCC [41].

In this study, we conducted a systematic review and meta-analysis to examine the prognostic value of GPRC5A in different types of human cancers from a collection of published results. Consistent with our hypothesis, we observed that high GPRC5A expression predicted poor OS, DFS, and EFS, in multiple cancer patients. To assess the specific relationship between the GPRC5A and the OS of each cancer type, subgroup analysis showed high expression of GPRC5A was significantly associated with poor prognosis in the majority of solid cancers studied such as pancreatic, gastric, prostate, hepatocellular and esophageal cancer, but no

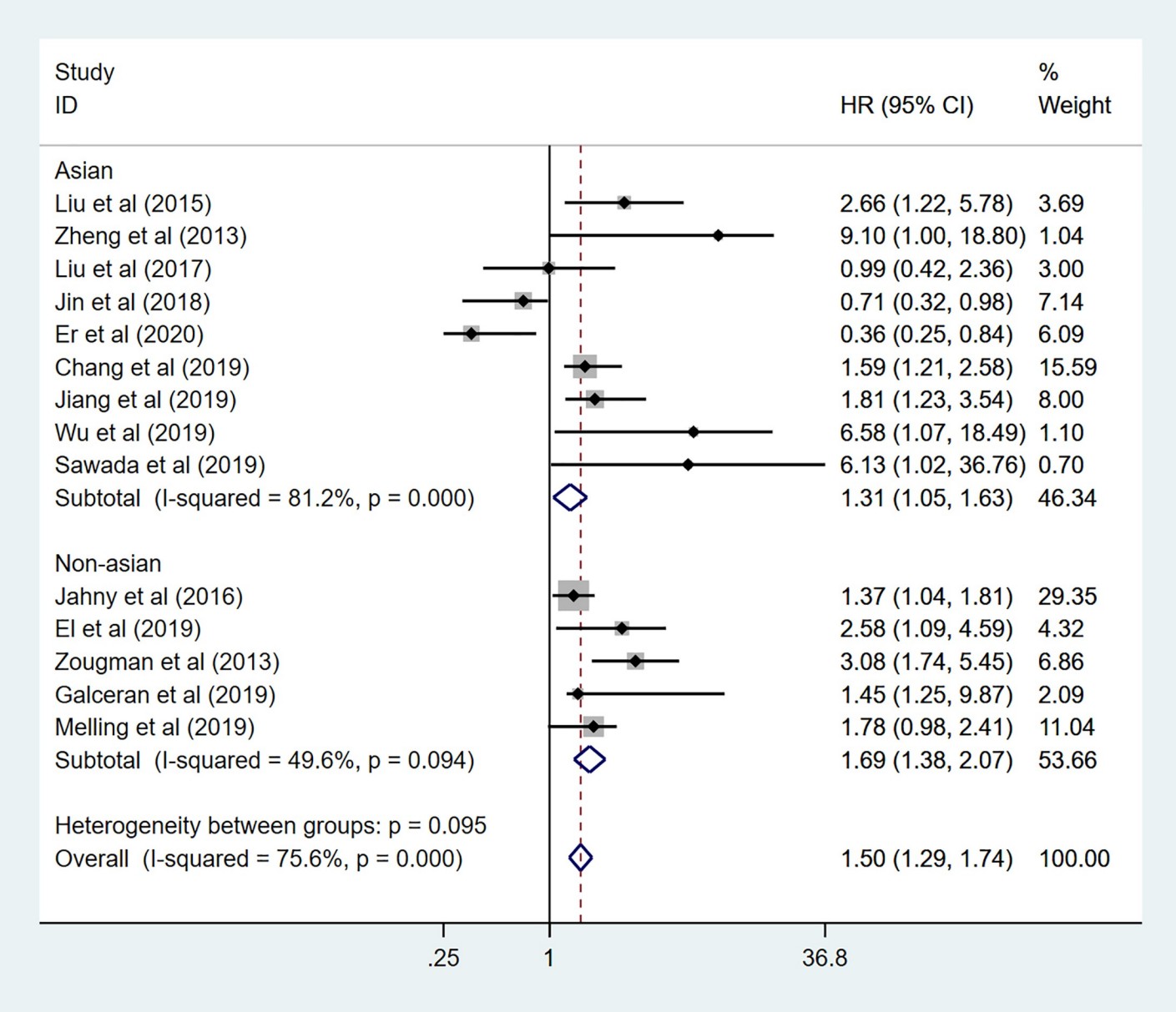

**Fig 8. Forest plot for the subgroup analysis.** Region.

significant effect was observed in colorectal and ovarian cancer. In lung, as well as head and neck squamous cancer, high GPRC5A expression was associated with favorable prognosis. These findings suggest that GPRC5A expression may have clinical potential as an independent prognostic indicator for some types of cancer patients; however, the application CPRC5A may be different based on the types of cancer.

The results of this study should be taken into consideration in the context of certain limitations. First, as only 15 studies were enrolled, the data were relatively insufficient to pool results by tumor type, which prevented us from obtaining more comprehensive results. Well-designed and large-scale cohort studies are needed to certify the clinical value of GPRC5A in multiple cancers. Second, all the studies enrolled in our meta-analysis were retrospective

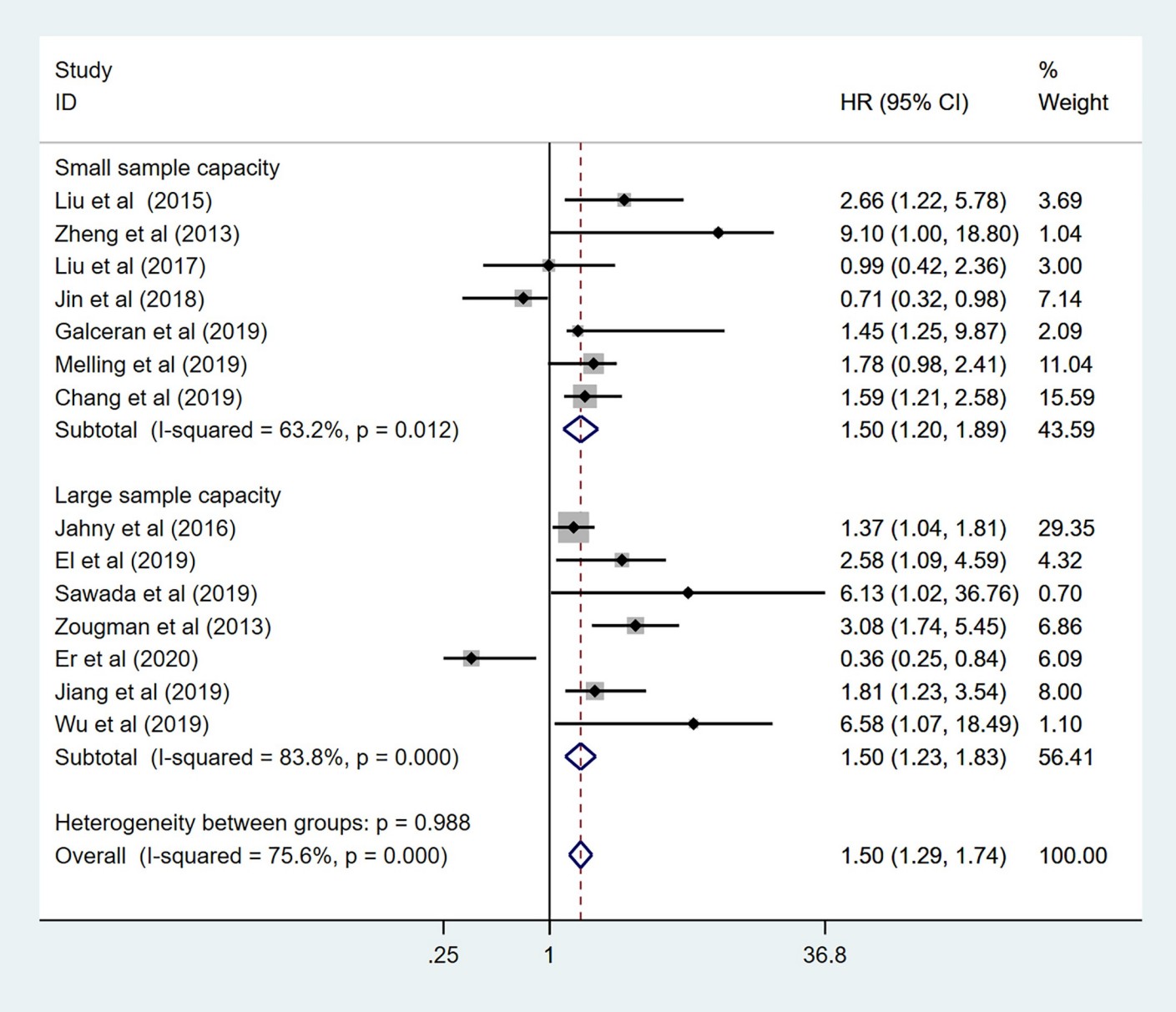

**Fig 9. Forest plot for the subgroup analysis.** Sample capacity.

articles. Most of these reports were not meant to explore the prognostic influence of GPRC5A. The accuracy of the collected data related to OS is unknown. HR and 95% CI values were not available for many of the studies, and extraction of the data from survival curves may have led to minor statistical errors. Third, heterogeneity existed in our study and may have been a result of the different cutoff values, tumor types, sample sources and follow-up periods across the studies. Additionally, we only included studies in English and Chinese, and records reported in other languages were omitted. Finally, although our results showed that the predictive value of GPRC5A varies significantly according to cancer type, we did not further study the mechanisms of this. Further studies will be needed to reveal novel insights into application of GPRC5A in cancer.

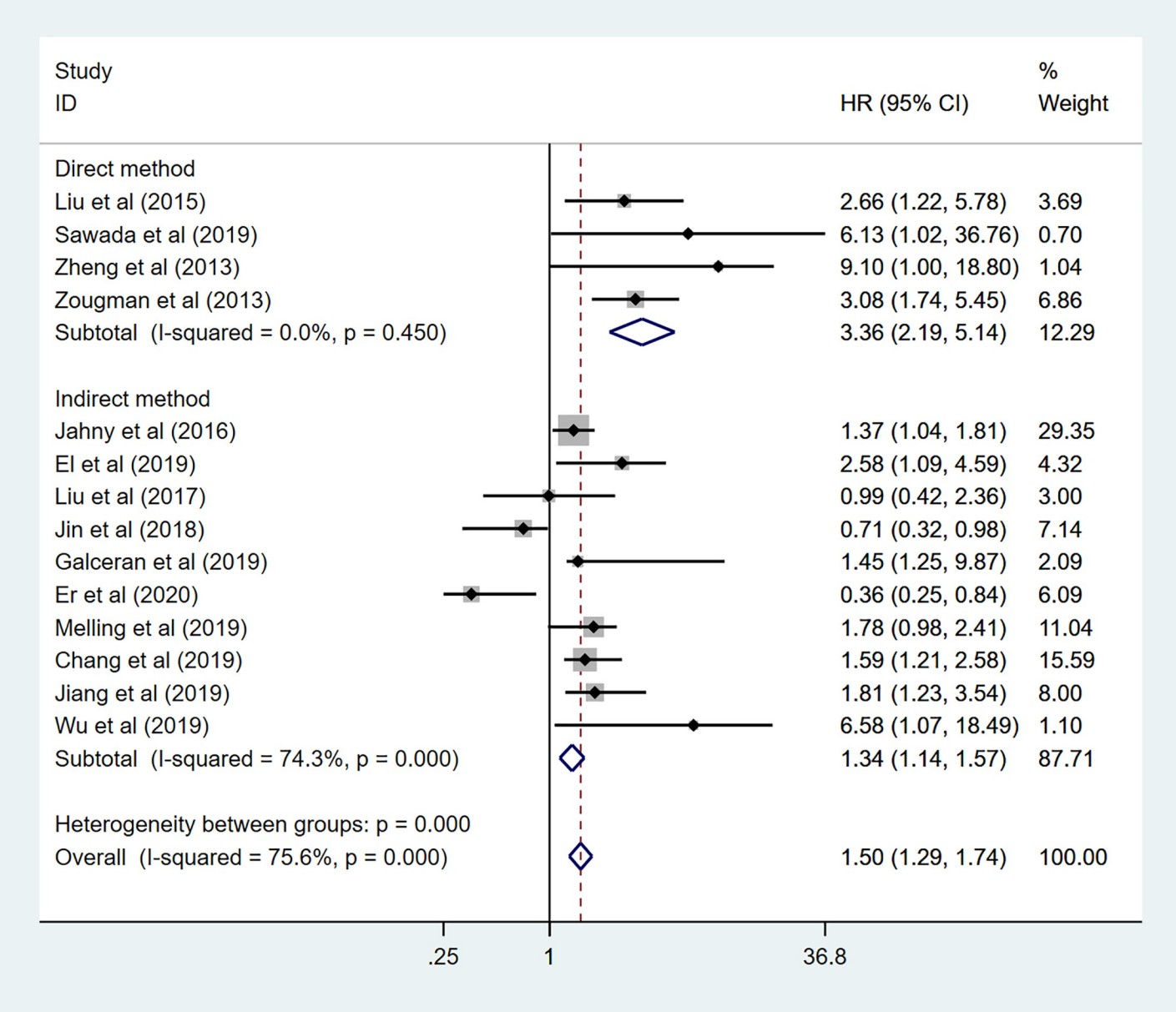

**Fig 10. Forest plot for the subgroup analysis.** HR obtained measurement.

## Conclusion

Although limitations were noted above, this was the first meta-analysis to systematically assess the prognostic value of GPRC5A in human cancer. We enrolled 15 recent studies covering nine different types of cancers reported between 2010 and 2018 for our meta-analysis. In summary, our results suggest that GPRC5A can be a promising candidate for predicting medical outcomes and used for accurate diagnosis, prognosis prediction for patients with cancer; however, the predictive value of GPRC5A varies significantly according to cancer type. Current knowledge of the exact mechanism of these processes is limited. Further studies focused on the cellular and molecular mechanisms will be necessary to reveal novel insights into application of GPRC5A in cancers.

**Table 3. Meta-analysis of the relationship between overexpressed GPRC5A and clinicopathological parameters.**

| | Studies(n) | Number of patients(n) | OR | LCI | UCI | Heterogeneity | | Model |
|---|---|---|---|---|---|---|---|---|
| | | | | | | I2 | P | |
| Sex | 7 | 1118 | 1.17 | 0.91 | 1.50 | 0.0% | 0.863 | Fixed effects |
| Age | 5 | 785 | 1.14 | 0.86 | 1.52 | 0.0% | 0.578 | Fixed effects |
| Tumor grade | 7 | 955 | 1.58 | 0.71 | 3.50 | 88.5% | 0.000 | Random effects |
| T stage | 5 | 957 | 1.83 | 1.15 | 2.92 | 61.3% | 0.035 | Random effects |
| Lymph node metastasis | 5 | 783 | 1.95 | 1.33 | 2.86 | 43.9% | 0.129 | Fixed effects |
| Metastasis | 4 | 602 | 1.40 | 0.59 | 3.32 | 65.0% | 0.036 | Random effects |

Note. LCI: lower confidence interval; OR: odds ratio; UCI: upper confidence interval.

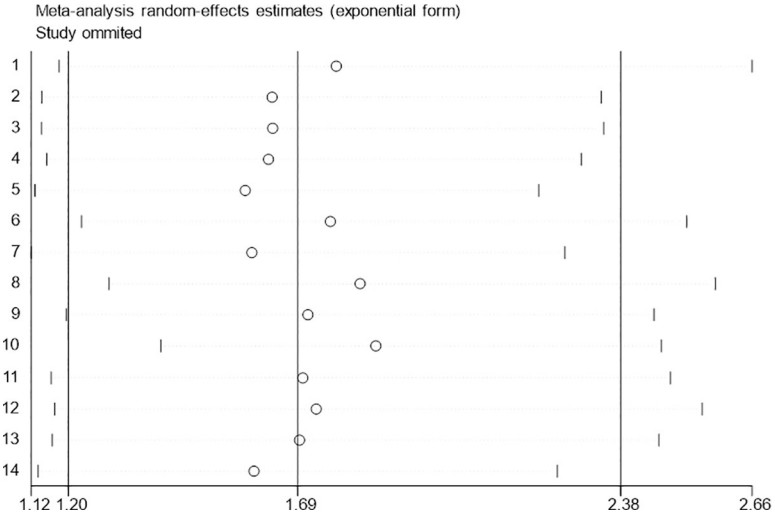

**Fig 11. Sensitivity analysis of the relationship between GPRC5A expression and OS.**

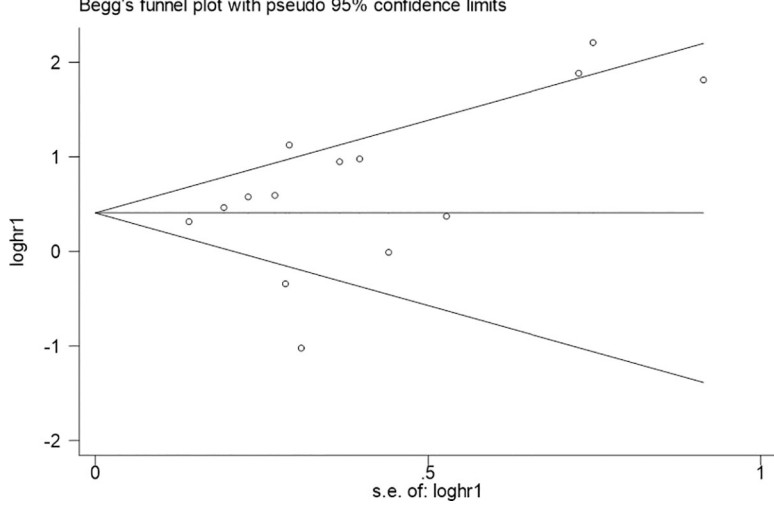

**Fig 12. Begg's funnel plot analysis of potential publication bias for OS.**

## Supporting information

**S1 Checklist.**
(DOC)

## Acknowledgments

We would like to thank the researchers and study participants for their contributions.

## Author Contributions

**Conceptualization:** Lu Dai.

**Data curation:** Lu Dai, Xiao Jin.

**Formal analysis:** Xiao Jin.

**Funding acquisition:** Xiao Jin.

**Methodology:** Lu Dai.

**Software:** Lu Dai.

**Supervision:** Zheng Liu.

**Writing – original draft:** Lu Dai.

**Writing – review & editing:** Lu Dai, Zheng Liu.

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
