## [Decision Letter · Decision Letter 0]

14 Jan 2021

PONE-D-20-28866

Prognostic and clinicopathological significance of GPRC5A in various cancers: a systematic review and meta-analysis

PLOS ONE

Dear Dr. liu,

Thank you for submitting your manuscript to PLOS ONE. After careful consideration, we feel that it has merit but does not fully meet PLOS ONE’s publication criteria as it currently stands. Therefore, we invite you to submit a revised version of the manuscript that addresses the points raised during the review process.

This manuscript was carefully reviewed by 2 experts, and both of them found several issues which need to be addressed before this manuscript becomes potentially accepted. For instance, reviewer 1 suggested revision of the main text including introduction and discussion. Reviewer 2 indicted English writing problems. Please respond to each of the reviewer comments.

We look forward to receiving your revised manuscript.

Kind regards,

Hiromu Suzuki, M.D., Ph.D.

Academic Editor

PLOS ONE

Journal Requirements:

2. Thank you for stating the following financial disclosure: 'NO'

4. Please include your tables as part of your main manuscript and remove the individual files. Please note that supplementary tables should remain as separate "supporting information" files.

5. Please include captions for your Supporting Information files at the end of your manuscript, and update any in-text citations to match accordingly. Please see our Supporting Information guidelines for more information: http://journals.plos.org/plosone/s/supporting-information

Reviewers' comments:

Reviewer's Responses to Questions

**Comments to the Author**

1. Is the manuscript technically sound, and do the data support the conclusions?

Reviewer #1: Yes

Reviewer #2: Yes

2. Has the statistical analysis been performed appropriately and rigorously? 

Reviewer #1: Yes

Reviewer #2: Yes

3. Have the authors made all data underlying the findings in their manuscript fully available?

Reviewer #1: Yes

Reviewer #2: Yes

4. Is the manuscript presented in an intelligible fashion and written in standard English?

Reviewer #1: No

Reviewer #2: No

5. Review Comments to the Author

Reviewer #1: The authors have worked with detail meta-analysis wiith large subgroup analysis

The following commets are need to be corrected

1. Please update the introduction part wiht recent comments

2. Discussion part should be dicussed based on subgroup analsyis as well

3. strength and limitations of the study should be included

4 I couldnt see PRISMA statement. Kindly include the info regarding PRISMA guidleines

5. details on publication bias should be discussed on methodology and results section in detail

6. results on quality assessment should be discussed in detail

Reviewer #2: English wording need to be improved

Discussion should be improved in terms of structure and concept and comparison with the literature. The current discussion section is not strong enough to support the findings.

6. PLOS authors have the option to publish the peer review history of their article (what does this mean?). If published, this will include your full peer review and any attached files.

Reviewer #1: **Yes: **Shanthi Sabarimurugan

Reviewer #2: No

---

## [Author Response · Author response to Decision Letter 0]

5 Feb 2021

The details of our edits in the manuscript are outlined below.

REVIEWER #1

The authors have worked with detail meta-analysis with large subgroup analysis

The following comments are needed to be corrected

• Our Response: We thank the reviewer for their comments and enthusiasm for our paper.

• 

Comment 1: Please update the introduction part with recent comments

• Our Response: We have updated the overall introduction section. Please see the revision in the second submission.

Comment 2: .Discussion part should be discussed based on subgroup analysis as well

• Our Response: We have added the discussion for subgroup analysis.

• From text: “To assess the specific relationship between the GPRC5A and the OS of each cancer type, subgroup analysis showed high expression of GPRC5A was significantly associated with poor prognosis in the majority of solid cancers studied such as pancreatic, gastric, prostate, hepatocellular and esophageal cancer, but no significant effect was observed in colorectal and ovarian cancer. In lung and head and neck squamous cancer, high GPRC5A expression was associated with favorable prognosis. These findings suggest that GPRC5A expression may have clinical potential as an independent prognostic indicator for some types of cancer patients; however, the application CPRC5A may be different based on the type of cancers.”

Comment 3. Strength and limitations of the study should be included

• Our Response: We did include both strength and limitations in the first submission and we also updated them in this revision. 

• From text: 

Strength: “Although limitations were noted above, this was the first meta‐analysis to systematically assess the prognostic value of GPRC5A in human cancers. We enrolled 15 recent studies covering nine different types of cancers reported between 2010 and 2018 for our meta-analysis. In summary, our results suggest that GPRC5A can be a promising candidate for predicting medical outcomes and used for accurate diagnosis, prognosis prediction for patients with cancer; however, the predictive value of GPRC5A varies significantly according to cancer type.”

Limitations: “The results of this study should be taken into consideration in the context of certain limitations. First, as only 15 studies were enrolled, the data were relatively insufficient to pool results by tumor type, which prevented us from obtaining more comprehensive results. Well-designed and large-scale cohort studies are needed to certify the clinical value of GPRC5A in multiple cancers. Second, all the studies enrolled in our meta-analysis were retrospective articles. Most of these reports were not meant to explore the prognostic influence of GPRC5A. The accuracy of the collected data related to OS is unknown. HR and 95% CI values were not available for many of the studies, and extraction of the data from survival curves may have led to minor statistical errors. Third, heterogeneity existed in our study and may have been a result of the different cutoff values, tumor types, sample sources and follow-up periods across the studies. Additionally, we only included studies in English and Chinese, and records reported in other languages were omitted. Finally, although our results showed that the predictive value of GPRC5A varies significantly according to cancer type, we did not further study the mechanisms of this. Further studies will be needed to reveal novel insights into application of GPRC5A in cancer..”

Comment 4: I couldn’t see PRISMA statement. Kindly include the info regarding PRISMA 

Guidelines

• Our Response: We did included PRISMA statement in the method section and we updated it in this revision.

• From text: 

“This study followed the PRISMA (Preferred Reporting Items for Systematic Reviews and Meta-Analyses) guidelines.”

Comment 5: details on publication bias should be discussed on methodology and results section in detail

• Our Response: We did include publication bias in the methods and result section in the first submission and we updated it in this revision.

• From text: 

“Publication bias was assessed by Begg’s and Egger’s tests.”

“Additionally, Begg’s funnel plots and Egger’s test showed no significant publication bias was found for OS (Fig. 6 Begg’sP ¼ 0.155 and Egger’s P ¼ 0.908).”

Comment 6: results on quality assessment should be discussed in detail

• Our Response: We did quality assessment in the first submission and we have added more details in this revision. 

• From text:

“Two investigators separately gained the information and data from primary publications. The specific information and data were as follows: The first author’s name, publication year, country, cancer type, time of sample collection, sample capacity, outcome measures, method of detection, and Cut-off value. For the clinically relevant factors, Age, Sex, differentiation, tumor invasion depth, lymph node metastasis, and distant metastasis were extracted. 

The Newcastle-Ottawa Scale (NOS) was also utilized to assess the quality of studies in the meta-analysis, which ranges from 0-9. A score of 5 or higher indicates strong evidence; a score from 4 to 5 (not included) indicates medium evidence, and a score below 4 indicates weak evidence. Studies with strong evidence (NOS score ≥ 5 points) were included in this study.”

REVIEWER #2

Comment 1: English wording need to be improved

• Our Response: We have improved the overall English wording of this paper. This paper has been reviewed by a native English speaker. Please see the revision in the second submission. 

Comment 2: Discussion should be improved in terms of structure and concept and comparison with the literature. The current discussion section is not strong enough to support the findings.

• Our Response: We have updated the overall discussion and added the discussion of subgroup analysis. Please see the revision in the second submission.

---

## [Decision Letter · Decision Letter 1]

10 Mar 2021

Prognostic and clinicopathological significance of GPRC5A in various cancers: a systematic review and meta-analysis

PONE-D-20-28866R1

Dear Dr. liu,

We’re pleased to inform you that your manuscript has been judged scientifically suitable for publication and will be formally accepted for publication once it meets all outstanding technical requirements.

Kind regards,

Hiromu Suzuki, M.D., Ph.D.

Academic Editor

PLOS ONE

Additional Editor Comments (optional):

Reviewers' comments:

Reviewer's Responses to Questions

**Comments to the Author**

1. If the authors have adequately addressed your comments raised in a previous round of review and you feel that this manuscript is now acceptable for publication, you may indicate that here to bypass the “Comments to the Author” section, enter your conflict of interest statement in the “Confidential to Editor” section, and submit your "Accept" recommendation.

Reviewer #1: All comments have been addressed

2. Is the manuscript technically sound, and do the data support the conclusions?

Reviewer #1: Yes

3. Has the statistical analysis been performed appropriately and rigorously? 

Reviewer #1: Yes

4. Have the authors made all data underlying the findings in their manuscript fully available?

Reviewer #1: Yes

5. Is the manuscript presented in an intelligible fashion and written in standard English?

Reviewer #1: Yes

6. Review Comments to the Author

Reviewer #1: (No Response)

7. PLOS authors have the option to publish the peer review history of their article (what does this mean?). If published, this will include your full peer review and any attached files.

Reviewer #1: **Yes: **Shanthi Sabarimurugan

---

## [Editor Report · Acceptance letter]

22 Mar 2021

PONE-D-20-28866R1 

Prognostic and Clinicopathological significance of GPRC5A in various cancers: A systematic review and meta-analysis 

Dear Dr. Liu:

I'm pleased to inform you that your manuscript has been deemed suitable for publication in PLOS ONE. Congratulations! Your manuscript is now with our production department. 

Kind regards, 

on behalf of

Dr. Hiromu Suzuki 

Academic Editor

PLOS ONE